# CRP Monitoring in Early Hospitalization: Implications for Predicting Outcomes in Patients with COVID-19

**DOI:** 10.3390/pathogens12111315

**Published:** 2023-11-04

**Authors:** Byron Avihai, Erin P. Sundel, Eileen Lee, Patricia J. Greenberg, Brennan P. Cook, Nicole J. Altomare, Tomohiro M. Ko, Angelo I. Chaia, Payal D. Parikh, Martin J. Blaser

**Affiliations:** 1Robert Wood Johnson Medical School, Rutgers University, Piscataway, NJ 08854, USA; esundel@bidmc.harvard.edu (E.P.S.); el412@rwjms.rutgers.edu (E.L.); bpc66@njms.rutgers.edu (B.P.C.); nicole.altomare@northwestern.edu (N.J.A.); tmk145@rwjms.rutgers.edu (T.M.K.); angelo.chaia@mountsinai.org (A.I.C.); parikhpd@rwjms.rutgers.edu (P.D.P.); 2Cancer Institute of New Jersey, Rutgers University, New Brunswick, NJ 08901, USA; 3Beth Israel Deaconess Medical Center, Boston, MA 02215, USA; 4Department of Biostatistics and Epidemiology, Rutgers School of Public Health, Piscataway, NJ 08854, USA; pjg134@sph.rutgers.edu; 5Northwestern Memorial Hospital, Chicago, IL 60611, USA; 6Mount Sinai Beth Israel, New York, NY 10003, USA; 7Center for Advanced Biotechnology and Medicine, Rutgers University, Piscataway, NJ 08854, USA

**Keywords:** C-reactive protein, COVID-19 hyperinflammatory syndromes, COVID-19 risk factors

## Abstract

Elevated C-reactive protein (CRP) levels have been associated with poorer COVID-19 outcomes. While baseline CRP levels are higher in women, obese individuals, and older adults, the relationship between CRP, sex, body mass index (BMI), age, and COVID-19 outcomes remains unknown. To investigate, we performed a retrospective analysis on 824 adult patients with COVID-19 admitted during the first pandemic wave, of whom 183 (22.2%) died. The maximum CRP value over the first five hospitalization days better predicted hospitalization outcome than the CRP level at admission, as a maximum CRP > 10 mg/dL independently quadrupled the risk of death (*p* < 0.001). Males (*p* < 0.001) and patients with a higher BMI (*p* = 0.001) had higher maximum CRP values, yet CRP levels did not impact their hospitalization outcome. While CRP levels did not statistically mediate any relation between sex, age, or BMI with clinical outcomes, age impacted the association between BMI and the risk of death. For patients 60 or over, a BMI < 25 kg/m^2^ increased the risk of death (*p* = 0.017), whereas the reverse was true for patients <60 (*p* = 0.030). Further impact of age on the association between BMI, CRP, and the risk of death could not be assessed due to a lack of statistical power but should be further investigated.

## 1. Introduction

Since the emergence of the novel coronavirus ARS-CoV-2 in 2019, researchers have investigated relationships between biological markers and COVID-19 outcomes. The acute phase reactant C-reactive protein (CRP), secreted by the liver in response to cytokines such as interleukin 6 (IL-6), rises within 6–8 h after the onset of acute viral or bacterial infections [1,2], aiding in innate immune responses [3]. Peak CRP levels occur about 48 h after an acute infection with a half-life of 19 h; levels may rise >1000-fold following acute infection or tissue injury [4]. CRP levels have been used for predicting COVID-19 disease severity [5,6,7,8,9,10,11]. Patients who were ultimately intubated, diagnosed with severe disease, or had worsened computer tomography (CT) severity scores often had elevated CRP levels [12,13,14,15,16].

These findings raise the question of whether CRP levels, early in the course of COVID-19, could be a marker in predicting illness severity, independent of other risk factors. Absolute CRP levels have been more predictive of patient outcomes than neutrophil counts or patient age [16], and CRP levels > 10 mg/dL also have been linked with COVID-19 hyperinflammatory syndrome and subsequent cytokine storm, harbingers of more severe outcomes [17]. Both the maximum (MAX) CRP value and the rate of CRP rise have also been predictive of COVID-19 severity [18,19,20]. However, prior studies have been heterogeneous, with variable CRP cut-off ranges and small sample sizes [19,20,21,22]. 

The relationship between the CRP level and patient age, obesity, and sex in COVID-19 also has not been resolved. In a study of the general population, people over 60 had higher CRP levels than younger adults [23]. However, whether elevated CRP levels are responsible for the association between older age and higher risk of severe COVID-19 [24,25,26,27,28,29,30] needs clarification. Similarly, obesity is a risk factor for COVID-19 severity and death [27,29,31,32,33], especially in middle-aged adults [34,35]. In the general population, obesity has also been associated with higher CRP levels [36,37,38]. In a study of 781 COVID-19 patients, obese persons had higher initial and MAX CRP levels than the non-obese; for each 1-standard deviation increased CRP level, patients were twice as likely to be admitted to the intensive care unit, have hypoxemic respiratory failure, or die [39]. Conversely, although CRP levels are generally lower at baseline in males than in females, male sex is another risk factor for severe COVID outcomes [27,40]. Thus, the relationship between body mass index (BMI), sex, and CRP in relation to risk for severe COVID-19 infection is complex and poorly understood.

To investigate these interactions, we performed a retrospective analysis evaluating the relationship between the CRP level and illness severity in adults who were admitted to a large New Jersey (USA) tertiary care hospital during the first COVID-19 wave. We studied the relationships between patient BMI, sex, age, and initial and MAX CRP levels to better predict COVID-19 disease severity and develop useful guidelines for clinicians.

## 2. Materials and Methods

### 2.1. Data Collection and Patient Population

We performed a retrospective analysis of 969 adults who were diagnosed as having COVID-19 when admitted to Rutgers Robert Wood Johnson University Hospital (RWJUH), a Level 1 trauma center in New Jersey, from 20 March 2020 and discharged on or by 31 May 2020. The study and informed consent waiver were approved by the Rutgers Institutional Review Board (protocol Pro20201234), and all methods were performed in accordance with the relevant guidelines and regulations. Of 969 adults, only the 824 non-pregnant patients who tested positive for SARS-CoV-2 infection by nasopharyngeal swab using polymerase chain reaction (PCR), and with both BMI and CRP values recorded within the first 5 days of admission were included in our study. A flow chart depicting patient inclusion and exclusion criteria is shown in Figure 1. Details about the entire cohort were recently published [41]. In order to maintain sufficient statistical power for any subsequent analysis, patients who suffered from diseases that can affect baseline CRP values, such as intercurrent infections, autoimmune disorders, or other chronic or acute conditions, were not excluded from the cohort.

We investigated two binary outcomes: death (deceased vs. survived) and clinical severity based on the American Thoracic Society Group (ATSG) criteria for outcomes of acute pneumonia (severe vs. non-severe). ATSG defines severe disease as admission to an ICU, invasive ventilation, or death and defines all other conditions as non-severe disease [42]. As per national guidelines at the time of the first wave of the COVID-19 pandemic [43], all patients who were admitted with a confirmed diagnosis of COVID-19 and who died were assumed to have died due to COVID-19. Data collected from the electronic medical records (EMR) included demographic information, insurance type, smoking history, vital signs, physical examination findings, comorbidities, laboratory tests, and presenting symptoms. Time 0 was defined as the time of the first entry of laboratory values in the EMR up to 24 h prior to hospital admission. From the medical records of the 824 patients, we extracted the CRP values recorded within 5 days of admission to the hospital, beginning up to 24 h prior to hospital admission, corresponding to a total interval of 5–6 days. We considered two different measures to evaluate the impact of CRP levels: the initial CRP, which corresponded to the first recorded CRP value, and the MAX CRP, which was defined as the maximum CRP value recorded over that time interval through hospital day 5. BMI values were calculated based on height and weight measured at hospital admission, and groups were divided into standard BMI tertiles: <25 kg/m^2^, 25.0–30.0 kg/m^2^, and >30 kg/m^2^, which correspond to underweight or healthy, overweight, and obese, respectively [44]. 

### 2.2. Statistical Analysis

We used descriptive statistics to examine characteristics of the COVID-19 patients, including sex, age, BMI, and CRP levels. Statistical significance throughout the study was defined as *p*-value < 0.05. To compare groups of patients within the cohort, we used the chi-square test without Yates’s correction for categorical variables (including hospitalization outcome, sex, BMI categories, and comorbidities) and the Wilcoxon rank-sum test for continuous variables (CRP distributions, initial and MAX CRP, age, lab values). To better evaluate the influence on hospitalization outcomes of continuous variables such as initial CRP, MAX CRP, and age on admission, these continuous variables were also assessed as binary categorical variables, respectively, initial CRP (>10 mg/dL), MAX CRP (>10 mg/dL) and age on admission (≥60 years old).

Linear regressions and multivariate logistic regressions were then used to assess the relationships between hospitalization outcomes and initial or MAX CRP as continuous and categorical (>10 mg/dL) variables. These regression models were further adjusted for well-established COVID-19 risk factors assessed as binary categorical variables such as sex, age (≥60 years old), and BMI (≥25 kg/m^2^) [17].

A mediator model following Baron and Kenny’s (1986) analysis steps was conducted to assess the relationship between CRP levels and well-established COVID-19 risk factors (BMI, age, sex) and disease outcomes [45]. If the first step of the mediation was not significant, but theoretical grounds for the relationship existed, we used Shrout and Bolger’s (2002) guidelines and proceeded step-wise in the mediation analysis [46].

## 3. Results

### 3.1. Characteristics of Study Population

Our analysis included 824 patients admitted to the hospital with COVID-19 who had a CRP determination within the first five days after admission and for whom a BMI value could be calculated. Based on the ATSG criteria, 545 (66.1%) of these patients developed non-severe COVID-19, and 279 (33.9%) patients had severe illness. Ultimately, 183 patients (22.2% of the cohort) died in the hospital. Patient comorbidities are reported in Appendix A. Other characteristics of the studied patients and their demographics relative to death outcome (Table 1) and ATSG classification (Appendix A) are shown. Laboratory values at admission relative to death outcome (Table 2) and ATSG classification (Appendix A) are also shown. On average, patients who died were older and had multiple differences in initial laboratory test results compared with patients who survived (Table 1 and Table 2).

First, we investigated whether initial CRP levels or MAX levels, obtained within 5 days of admission, were predictive of clinical outcomes. The distributions of initial and MAX CRP levels within the first 5 days of admission relative to death outcome and ATSG classification are shown in Figure 2 and Appendix A, respectively. CRP values obtained within the first 30 days of hospital admission relative to death outcome and ATSG classification are shown in Figure 3 and Appendix A, respectively. For all subsequent analyses, with one notable exception, findings that were statistically significant for death as the primary outcome were also statistically significant for the ATSG severity criteria as the primary outcome, and vice versa. The unique exception pertains to the interaction between BMI and age, which was only significant for death as a primary outcome. Therefore, unless specified, hospitalization outcome encompasses both ATSG severity and risk of death. For the sake of simplification, in the remaining manuscript, all statistics reported will pertain to death as the primary outcome. For statistics pertaining to ATSG severity, refer to Appendix A.

### 3.2. Relationship of CRP Levels to COVID-19 Patients’ Death

Patients in the deceased group had significantly (*p* < 0.001) higher initial (12.60 [7.46–21.24] mg/dL) and MAX (21.86 [13.16–30.79] mg/dL) CRP values compared to subjects who survived (with initial and MAX CRP values 9.52 [4.82–16.43] and 13.14 [6.58–22.65] mg/dL, respectively) (Figure 2, Table 2). Each 1 mg/dL incremental increase of initial CRP level obtained within the first 24 h of hospital admission was associated with an increase in the likelihood of death of 1.005 (95% CI: 1.002, 1.008; *p* < 0.001). Patients with initial CRP levels > 10 mg/dL were 1.62 times (95% CI: 1.17, 2.23; *p* = 0.004) more likely to die than those with initial values ≤ 10 mg/dL. Similarly, patients were 1.009 times (95% CI: 1.006, 1.011; *p* < 0.001) more likely to die for each 1 mg/dL increase of MAX CRP. Patients with MAX CRP levels > 10 mg/dL were 3.99 times (95% CI: 2.59, 6.39; *p* < 0.001) more likely to die than those with MAX CRP levels ≤ 10 mg/dL (Table 3).

MAX CRP levels were a better predictor of risk of death than initial CRP levels, as patients with MAX CRP levels > 10 mg/dL vs. ≤ 10 mg/dL were more than twice as likely to die compared to patients with initial CRP levels > 10 vs. ≤ 10 mg/dL (odds ratio of 3.99 vs. 1.62). This can be due to maximum CRP levels over a time interval (5 days) being less prone to CRP fluctuations than a single initial measure. However, using MAX CRP over the first 5 days of hospitalization is appropriate given that 71% of patients in our cohort spent at least 5 days in the hospital (median 8 [IQR: 4–12] days). Patients who died stayed longer (10 [IQR: 6–17] days) than patients who survived (7 [IQR: 4–11] days; *p* < 0.001), with 84.2% of patients who died staying at least 5 days, vs. 67.6% of patients who survived (*p* < 0.001). Therefore, regardless of hospitalization outcome, most patients had sufficiently long hospitalizations for a maximum CRP value over 5 days to be relevant. For subsequent investigations, we used MAX CRP to identify risk factors associated with increased CRP levels.

### 3.3. Relationship between MAX CRP Values with Age and COVID-19 Outcomes

We used logistic regressions to investigate the relationship between age and risk of death. In a univariate assessment, patients ≥ 60 years old had 4.20 (95% CI: 2.83, 6.41; *p* < 0.001) greater odds of dying compared to patients < 60. However, patients ≥ 60 years old were not at an increased risk of having MAX CRP levels > 10 mg/dL compared to patients < 60 years [OR: 0.93 (95% CI: 0.70, 1.25); *p* = 0.65] (Table 3). To investigate the relationship between age and MAX CRP with hospitalization outcome, we combined these variables into a two-way multivariate logistic regression with age ≥ 60 years and MAX CRP > 10 mg/dL as categorical variables. There was no significant interaction between age ≥ 60 years and MAX CRP > 10 mg/dL [OR: 0.48 (95% CI: 0.07, 1.82); *p* = 0.35] in predicting the likelihood of death (Table 3). In total, these results indicate that while adults ≥ 60 were more likely to die, it was likely not causally related to an increase in CRP-induced activities.

### 3.4. Relationship between MAX CRP Values, Sex, and COVID-19 Outcomes

In our COVID-19 patient population, males had higher MAX CRP values (median 16.87 [IQR: 8.35–26.97] mg/dL) than females (12.66 [IQR: 6.42–20.77] mg/dL [*p* < 0.001]). Males also were significantly more likely to have MAX CRP levels > 10 mg/dL compared to females [OR: 1.44 (95% CI: 1.07, 1.94); *p* = 0.015]. However, males were no more likely than females to die [OR: 0.91 (95% CI: 0.65, 1.27); *p* = 0.57] (Table 3). Nevertheless, both males who died and males who survived had higher MAX CRP values (respectively, 25.37 [IQR: 15.22–32.62] and 14.96 [IQR: 7.84–25.08]) compared to their female counterparts (17.31 [IQR: 11.51–25.64] [*p* = 0.002] and 11.59 [IQR: 5.39–19.00] [*p* < 0.001]) (Appendix A).

We also performed two-way multivariate logistic regressions with sex and MAX CRP > 10 mg/dL and age ≥ 60 years as binary categorical variables. There was no significant interaction between sex and MAX CRP values > 10 mg/dL associated with the likelihood of death [OR: 1.21 (95% CI: 0.48, 3.02); *p* = 0.67]. MAX CRP > 10 mg/dL also was not a statistical mediator between sex and risk of dying. There also was no significant interaction between age as a categorical variable (<60 vs. ≥60 years old) and sex for predicting risk of death [OR: 0.60 (95% CI: 0.20, 1.55); *p* = 0.32] (Table 3). From these data, we conclude that even though males were more likely than females to have higher MAX CRP, they were no more likely to die than females, and elevated CRP levels were not related to their hospitalization outcomes.

### 3.5. Relationship between MAX CRP Values, Body Mass Index, Age and COVID-19 Outcomes

We studied the interaction between BMI and MAX CRP in relation to hospitalization outcomes. There was no difference in the proportion of obese patients who died (21.3%) compared to overweight patients (20.7%, *p* = 0.844). There also was no difference in MAX CRP overall distributions between obese (17.19 [IQR: 9.17–26.33]) and overweight patients (14.89 [IQR: 7.85–26.90] mg/dL; *p* = 0.23) across the cohort, nor within the sub-cohorts of patients who died (22.09 [IQR: 14.38–34.62] vs. 22.01 [IQR: 12.83–30.95]; *p* = 0.44) or survived (15.31 [IQR: 8.76–22.67] vs. 13.38 [IQR: 6.32–25.33]; *p* = 0.27) (Appendix A). Since there was no significant difference between obese and overweight groups, we categorized patients into two BMI groups for any subsequent analysis: BMI ≥ 25 kg/m^2^ (overweight or obese) vs. BMI < 25 kg/m^2^ (normal or underweight).

While patients with a BMI ≥ 25 kg/m^2^ were no more likely to die than patients with a BMI < 25 kg/m^2^ [OR: 0.81 (95% CI: 0.57, 1.14); *p* = 0.22], they were 1.65 times more likely to have a MAX CRP value > 10 mg/dL (95% CI: 1.21, 2.24; *p* = 0.0014). Nevertheless, there was no significant interaction, as assessed by two-way multivariate logistic regression, between BMI (≥25 vs. <25 kg/m^2^) and MAX CRP values (>10 vs. ≤10 mg/dL) for predicting the risk of dying [OR: 1.03 (95% CI: 0.41, 2.59); *p* = 0.95] (Table 3). Furthermore, MAX CRP as a binary categorical value did not statistically mediate the relationship between BMI and hospitalization outcome.

We then hypothesized that age might mask the relationship between BMI, CRP, and hospitalization outcome. Indeed, a two-way multivariate logistic regression model found a significant interaction between BMI and age in predicting the risk of death. Among patients ≥ 60 years old, a BMI ≥ 25 kg/m^2^ decreased the risk of death (normal or underweight) [OR: 0.16 (95% CI: 0.02, 0.59), *p* = 0.017], whereas, among patients < 60 years old, a BMI ≥ 25 kg/m^2^ increased risk of death [OR: 4.99 (95% CI: 1.46, 31.27); *p* = 0.030] (Table 3). The over-representation of older patients among patients with a BMI below 25 kg/m^2^ (70.6% were ≥60) compared to patients with a BMI of 25 kg/m^2^ or above (53.0% were ≥60; *p* < 0.001) masked the predictive power of BMI as it relates to the risk of death. However, the interaction between BMI and age was not significant when using the ATSG criteria as the hospitalization outcome [OR: 0.52 (95% CI: 0.25, 1.06), *p* = 0.078] (Appendix A). This led us to use death as the primary hospitalization outcome for this study.

Since age confounded the impact of BMI on predicting the risk of death, we hypothesized that age also may confound the interaction between BMI and CRP in predicting outcomes. To address that question, we conducted a three-way multivariate logistic regression to assess the interaction of age, BMI, and CRP in relation to the risk of death, as well as a two-way multivariate logistic regression stratified by age to assess the interaction between BMI and CRP, all as binary categorical variables. However, no relationships were significant (*p* = 0.98).

In summary, these data indicate that adults ≥60 with a lower BMI and adults < 60 with a higher BMI were at greater risk of dying. However, due to limited statistical power, our analysis could not determine whether higher MAX CRP levels, which were detected in patients of higher BMI, mediated the impact of BMI and age on clinical outcomes.

## 4. Discussion

To assess the relationship between CRP, risk factors, and COVID-19 outcomes, two binary clinical outcomes were investigated: death (deceased vs. survived) and clinical severity based on the ATSG criteria (severe vs. non-severe) [42]. With one notable exception, we found that significant findings for death as the primary outcome were also significant for ATSG severity as the primary outcome and vice versa. Therefore, when not mentioned otherwise, hospitalization outcome encompasses both ATSG severity and risk of death.

In patients diagnosed with SARS-CoV-2 infection, higher CRP levels have been associated with severe pulmonary pathology and with worse overall outcomes [12,13,14,15,16,18,19,20,21,22]. Moreover, elevated CRP has been used as a predictor of COVID-19 severity and mortality [5,6,7,8,9,10,11]. However, patients with SARS-CoV-2 present for medical attention at differing points in their illness [47]. In our study, we took advantage of a SARS-CoV-2-naïve population presenting with acute illness in the first COVID-19 wave in the USA. To standardize the analysis and optimize clinical utility, we examined CRP determinations in two ways: CRP at hospital admission and the MAX CRP value over the first 5 days of hospitalization. Of the two, we found MAX CRP to be more predictive of hospitalization outcomes. Patients with MAX CRP levels > 10 mg/dL were 3.99 times more likely to die compared to patients with MAX CRP levels ≤ 10 mg/dL, whereas patients with initial CRP values > 10 mg/dL were only 1.62 times more likely to die than patients with initial CRP values ≤ 10 mg/dL. Indeed, initial levels are highly dependent on when patients seek medical care for their illness and on CRP fluctuations. Therefore, trending the MAX CRP levels over the first 5 days may be useful to clinicians. After 5 days, the dichotomy in the courses between benign and more severe clinical outcomes becomes more clear, thus decreasing the need for additional predictive biomarkers.

Three known risk factors for COVID-19 severity were assessed in conjunction with CRP in our analysis: patient sex, age, and BMI. In the general population, CRP levels are higher in females than males, even when stratified by BMI or adjusted for age, ethnicity, diabetes, hypertension, smoking, and alcohol consumption [48,49]. However, males with COVID-19 from our cohort had significantly higher CRP levels compared to females (Appendix A), even though they were no more likely to die than their female counterparts, providing evidence that their higher CRP levels were not related to their hospitalization outcomes.

Although increased age is generally associated with more severe SARS-CoV-2 infection [24,25,26], there has been little information about the relationship between increased age and CRP levels in such patients. In a study of the general population, patients >60 years old had higher CRP levels than younger adults [23]. While patients ≥60 were at significantly higher risk of poorer hospitalization outcomes in our study, they did not have higher maximum CRP levels compared to patients <60. This finding suggests that the increased mortality in the elderly is not due to a corresponding increase in CRP levels and that mechanisms other than CRP-mediated inflammation worsen outcomes in the elderly.

In the general population, CRP and BMI levels are positively correlated for both sexes regardless of race and ethnicity, consistent with obesity creating a pro-inflammatory state [37,50]. In our study of COVID-19 patients, MAX CRP levels were higher in those overweight or obese compared to normal or underweight patients (Appendix A), consistent with prior findings [39]. However, as with age, our analyses do not indicate that CRP is a biological mediator between BMI and hospitalization outcome.

Adipose tissue produces IL-6, which stimulates CRP production, and obese patients also have increased circulating levels of NF-kB, TNF-a, and IL-1 [51]. IL-6 stimulates CRP release via both the “classical signaling” and “trans-signaling” pathways. Adipocytes mainly use the “trans-signaling” pathway to stimulate IL-6 and, therefore, CRP [52]. However, a longitudinal Swedish cohort study found that the “classical” IL-6 signaling pathway dominated in severe COVID-19 over the “trans-signaling” pathway [53]. Although higher BMI has been clearly associated with worse COVID-19 outcomes [31,32], it is unknown whether the risk is due to increased “classical-signaling” activity, independent of adipose tissue-induced CRP release, or other inflammatory cascades.

Alternatively, the lack of CRP mediation between BMI and hospitalization outcome in our study may reflect patient age confounding the analyses. Obesity is less prevalent in older persons [54], and our elderly cohort (63 [IQR: 51–75] years) followed that trend: 70.6% of patients with a BMI < 25 kg/m^2^ were ≥60, vs. 53.0% of patients with a BMI ≥ 25 kg/m^2^. Among elderly patients within our cohort, those with a lower BMI were more likely to die during hospitalization, while those with a higher BMI were more likely to die for patients < 60. This interaction between age and BMI was the only one that was significant for death as the primary outcome but was not significant when predicting clinical severity according to the ATSG criteria. Although age could confound the BMI and CRP interactions related to hospitalization outcomes, our study lacked sufficient statistical power to test that hypothesis. Future studies with larger sample sizes should investigate the three-way interaction between BMI, CRP, and age in relation to COVID-19 patient outcomes.

Limitations of this study include that the analysis was conducted at a single hospital, limiting its generalizability. Furthermore, the retrospective design does not allow direct testing of hypotheses about causal relationships. Waist circumference is associated with elevated CRP levels, independent of BMI, and better assesses visceral obesity [55]. However, the retrospective nature of the study limited us to the use of BMI, which does not provide an accurate estimation of body fat composition, especially for visceral adipose tissue [56]. The absence of CRP mediation between BMI and hospitalization outcome could be explained by the advanced age of the cohort, where BMI may fail to represent visceral adiposity, an important correlate with respiratory failure in COVID-19 patients [57]. CRP levels also were not always ascertained daily, and patients with more severe initial symptoms may have been tested more frequently. However, by examining MAX CRP levels across the first 5 days, we sought to limit the effects of differences in sampling intensity. Patients who suffered from autoimmune conditions and other diseases that could affect baseline CRP values were also not excluded from the analysis to maintain sufficient statistical power. Future analysis of larger patient cohorts could help properly study the many confounding variables.

## 5. Conclusions

In conclusion, the maximum CRP value evaluated over the first 5 days of hospital admission can be used as a rapid and inexpensive approach for clinicians to efficiently triage patients to predict COVID-19 severity, independent of known risk factors such as patient age, sex, and BMI. Such triage can reduce unnecessary and toxic therapies in patients at low risk for poor outcomes. While CRP did not appear to mediate any relation between either sex, age, or BMI and clinical outcome, age had a major impact on the association between BMI and risk of death. Indeed, for patients who were 60 or above, having a BMI below 25 kg/m^2^ was a risk factor for dying, whereas for patients under 60, having an elevated BMI (25 kg/m^2^ or above) was associated with an increased risk of dying. Further studies on larger cohorts should be conducted to assess the impact of age on the association between BMI, CRP, and risk of death due to COVID-19.

## Figures and Tables

**Figure 1 pathogens-12-01315-f001:**
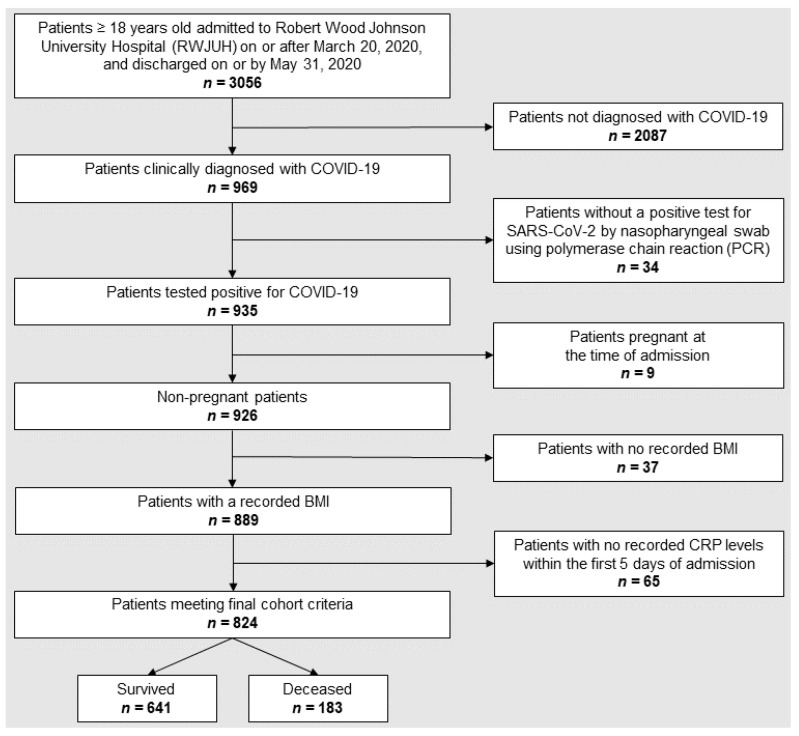
Flow chart depicting patient selection and exclusion criteria.

**Figure 2 pathogens-12-01315-f002:**
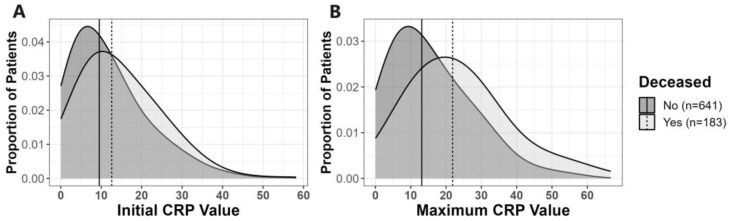
Distribution of initial and maximum CRP values among survivors and deceased. For each graph, the dashed black line represents the median of the corresponding CRP values among deceased patients, and the solid black lines represent the median of the corresponding CRP values among survivors. Panels: (**A**) distribution of initial CRP. For deceased patients, the median initial CRP was 12.60 (IQR: 7.46–21.24) vs. 9.52 (IQR: 4.82–16.43) mg/dL for those who survived (*p* < 0.001); (**B**) distribution of maximum CRP. For deceased patients, the median MAX CRP was 21.86 (IQR: 13.16–30.79) vs. 13.14 (IQR: 6.58–22.65) mg/dL for those who survived (*p* < 0.001).

**Figure 3 pathogens-12-01315-f003:**
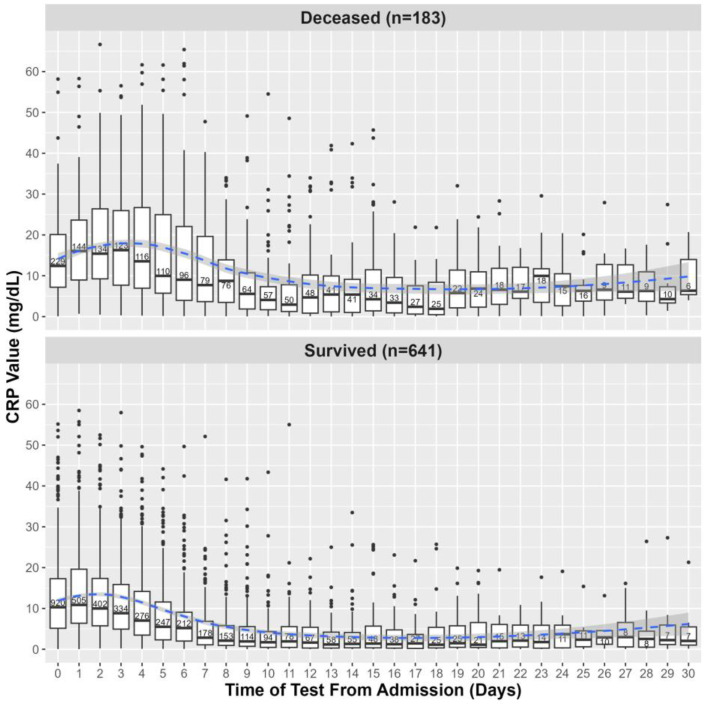
CRP levels within the first 30 days of hospital admission among survivors and deceased. The values shown on day 0 are defined by CRP values collected within the 24 h preceding hospital admission. Each boxplot shows the median and IQR of CRP values; points are outliers, and in each box is written the number of CRP values obtained on the corresponding day. The blue line represents a local regression plotting CRP values over time, and the shaded area corresponds to the 95% confidence interval of the regression.

**Table 1 pathogens-12-01315-t001:** Characteristics of the cohort based on outcomes in 824 hospitalized patients with COVID-19.

Demographics	Overall (*n* = 824)	Survived (*n* = 641, 77.8%)	Deceased (*n* = 183, 22.2%)	*p*-Value ^+^
**Age on Admission**	63 (IQR: 51–75) *	60 (IQR: 49–72)	72 (IQR: 63.5–83.5)	<0.001
**LOS (days)**	8 (IQR: 4–12)	7 (IQR: 4–11)	10 (IQR: 6–17)	<0.001
Number of patients with LOS ≥ 5 days	587 (71.2%) **	433 (67.6%)	154 (84.2)	<0.001
**Sex**				0.573
Female	314 (38.1%)	241 (37.6%)	73 (39.9%)	
Male	510 (61.9%)	400 (62.4%)	110 (60.1%)	
**Ethnicity**				<0.001
African American	115 (14.0%)	95 (14.8%)	20 (10.9%)	
Asian	57 (6.8%)	43 (6.7%)	14 (7.7%)	
Hispanic	276 (33.5%)	235 (36.7%)	41 (22.4%)	
Other	26 (3.2%)	20 (3.1%)	6 (3.3%)	
South Asian	46 (5.6%)	38 (5.9%)	8 (4.4%)	
White	304 (36.9%)	210 (32.8%)	84 (51.4%)	
**BMI Group**				0.463
Underweight or Normal	262 (31.8%)	197 (30.7%)	65 (35.6%)	
Overweight	276 (33.5%)	219 (34.2%)	57 (31.1%)	
Obese	286 (34.7%)	225 (35.1%)	61 (33.3%)	
**Comorbidity**				
Obesity	288 (35.0%)	226 (35.3%)	62 (33.9%)	0.730
Hypertension	523 (63.5%)	389 (60.7%)	134 (73.2%)	0.002
CAD or MI	161 (19.5%)	108 (16.8%)	53 (29.0%)	<0.001
Diabetes	330 (40.0%)	252 (39.3%)	78 (42.6%)	0.420
Chronic Kidney Disease	91 (11.0%)	61 (9.5%)	30 (16.4%)	0.009
Chronic Liver Disease	23 (2.8%)	15 (2.3%)	8 (4.4%)	0.141
Autoimmune Condition	41 (5.0%)	30 (4.7%)	11 (6.0%)	0.465
Asthma	69 (8.4%)	59 (9.2%)	10 (5.5%)	0.107
COPD	71 (8.6%)	48 (7.5%)	23 (12.6%)	0.031
Pulmonary—Other	55 (6.7%)	41 (6.4%)	14 (7.7%)	0.549
Malignancy (history of)	110 (13.3%)	70 (10.9%)	40 (21.9%)	<0.001
**ATSG Category**				<0.001
Non-Severe	545 (66.1%)	545 (85.0%)	0 (0%)	
Severe	279 (33.9%)	96 (15.0%)	183 (100%)	

LOS: length of stay; CAD: coronary artery disease; MI: myocardial infarction; COPD: chronic obstructive pulmonary disease. For binning methods of comorbidities, refer to Appendix A. * Median (IQR) for continuous variables. ** Counts (percent within the group) for categorical variables. ^+^ Statistical tests used to compare deceased and survivor groups: Wilcoxon rank-sum test for continuous variables, chi-square test without Yates’s correction for categorical variables.

**Table 2 pathogens-12-01315-t002:** Results of laboratory tests obtained early in the hospitalization of 824 COVID-19 patients in relation to clinical outcomes.

Laboratory Values *[Median (IQR)]*	Overall (*n* = 824)	Survived (*n* = 641, 77.8%)	Deceased (*n* = 183, 22.2%)	*p*-Value *	Reference Range
**C-Reactive Protein**					
Max CRP	15.02 (7.71–25.24)	13.14 (6.58–22.65)	21.86 (13.16–30.79)	<0.001	<0.07 mg/dL
On Admission	10.32 (5.26–17.63)	9.52 (4.82–16.43)	12.60 (7.46–21.24)	<0.001	<0.07 mg/dL
**Admission values**					
[Eosinophil]	0.00 (0.00–0.03)	0.01 (0.00–0.03)	0.00 (0.00–0.01)	<0.001	0.03–0.27 × 10^3^/μL
[Lymphocyte]	0.83 (0.59–1.21)	0.87 (0.64–1.25)	0.73 (0.51–1.06)	<0.001	1.16–3.18 × 10^3^/μL
[Neutrophil]	6.10 (4.34–8.83)	5.99 (4.23–8.33)	6.84 (4.74–9.82)	0.009	2.00–7.15 × 10^3^/μL
BUN	18 (11–32)	15 (11–26)	30 (20–49)	<0.001	6–23 mg/dL
D-Dimer	1091 (636–1932)	1023 (570–1813)	1207 (861–3342)	<0.001	0–500 ng/mL
Ferritin	779 (410–1475)	735 (396–1376)	1037 (453–1786)	0.004	20–335 ng/mL
INR	1.17 (1.08–1.29)	1.16 (1.08–1.28)	1.19 (1.08–1.34)	0.035	0.8–1.1
[WBC]	7.9 (5.9–10.7)	7.7 (5.8–10.3)	8.4 (6.1–11.4)	0.026	4.0–10.0 × 10^3^/μL

Max CRP: maximum CRP value within 5 days of admission; [] indicate absolute cell counts. * Wilcoxon rank-sum test used to compare deceased and survivor groups.

**Table 3 pathogens-12-01315-t003:** Results of linear regression and logistic regression models of sex, BMI, age, and CRP in relation to the likelihood of death.

Regression Models	Odds Ratio (95% CI)	*p*-Value
**CRP**		
Initial CRP (continuous) → Death	1.005 (1.002, 1.008)	<0.001
Initial CRP (>10 mg/dL) → Death	1.62 (1.17, 2.23)	0.004
MAX CRP (continuous) → Death	1.009 (1.006, 1.011)	<0.001
MAX CRP (>10 mg/dL) → Death	3.99 (2.59, 6.39)	<0.001
**Age**		
Age (≥60 years old) → Death	4.20 (2.83, 6.41)	<0.001
Age (≥60 years old) → MAX CRP (>10 mg/dL)	0.93 (0.70, 1.25)	0.65
Age (≥60 years old) × MAX CRP (>10 mg/dL) → Death *	0.48 (0.07, 1.82)	0.35
**Sex**		
Sex (M vs. F) → Death	0.91 (0.65, 1.27)	0.57
Sex (M vs. F) → MAX CRP (>10 mg/dL)	1.44 (1.07, 1.94)	0.015
Sex (M vs. F) × MAX CRP (>10 mg/dL) → Death	1.21 (0.48, 3.02)	0.67
Sex (M vs. F) × Age (≥60 years old) → Death	0.60 (0.20, 1.55)	0.32
**BMI**		
BMI (≥25 kg/m^2^) → Death	0.81 (0.57, 1.14)	0.22
BMI (≥25 kg/m^2^) → MAX CRP (>10 mg/dL)	1.65 (1.21, 2.24)	0.0014
BMI (≥25 kg/m^2^) × MAX CRP (> 10 mg/dL) → Death	1.03 (0.41, 2.59)	0.95
BMI (≥25 kg/m^2^) × Age (≥60 years old) → Death	0.16 (0.02, 0.59)	0.017
*Among Age < 60 years old:* BMI (≥25 kg/m^2^) → Death	4.99 (1.46, 31.27)	0.030

* Two-way multivariate logistic regression models are symbolized by “×” between the two variables.

## Data Availability

The data presented in this study are available for IRB-approved requests from the corresponding author. The dataset is not publicly available to protect patient privacy, as it contains protected health information.

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
