# Peer review of "CRP Monitoring in Early Hospitalization: Implications for Predicting Outcomes in Patients with COVID-19"

_pathogens, 2023, doi:10.3390/pathogens12111315_

Round 1

Reviewer 1 Report

Comments and Suggestions for Authors

Dear authors,

I read with particular interest his manuscript entitled “CRP monitoring in early hospitalization: implications for predicting outcomes in patients with COVID-19”. The authors tried to investigate the relationship between CRP level and illness severity in patients with COVID-19 who were hospitalized.

This paper is well written, and the study is well detailed, but I would like the authors to clarify some aspects.

Abbreviations should be defined at the first occurrence and introduced where multiple use is made.

Materials and methods

Page 2, line 74: Refer to Figure S1, which details the cohort selection. This figure could be included as the main material. Tables 1 and 2 would be results and do not provide information on case selection.

Specify in this section when (at what time after hospitalization) measurements of CRP, MAX CRP, and BMI levels were taken. The initial CRP levels were recorded within the first 5 days of admission, but the MAX CRP levels were taken on what day and when were they considered the maximum level?

Page 2, line 80: clinical severity of COVID-19?

What has been used for statistical analysis? What level of significance has been established?

Results

Page 3, line 122: “more severe COVID-19 disease…”

Page 3, line 123: “with those” is repeated.

Table 1 “Age on admission” No disaggregation of ages is given, indicating that it is a median of the age in each group studied.

Page 3, line 122: “On average, those with more severe disease were older…” Table 1 shows that the deceased patients were older, but this does not mean they have a more severe COVID-19 disease.

Page 6, line 160: “3.2. Relationship of CRP levels to COVID-19 patients death”

Page 8, line 248: It should be indicated that both groups, BMI< 25 kg/m and BMI≥ 25 kg/m, were ≥ 60 years old.

Discussion

Page 8, lines 270-273: These sentences help to understand how the CRP levels were examined; something similar should appear in materials and methods to clarify the methodology, as I have indicated in previous comments.

Conclusions

Page 9, line 341: Maximum CRP values were measured over the five days of hospital admission, right? Clarify

It would be interesting to add other risk factors, such as underlying diseases, immunosuppression, and cormobilities... that could contribute to the severity of COVID-19 and modulate the CRP levels being analyzed.

Are the group of deceased persons deceased by COVID-19, or did they have COVID-19 but died from other causes?

Thank you

Reviewer 2 Report

Comments and Suggestions for Authors

The current article titled “CRP monitoring in early hospitalization: implications for CRP monitoring in early hospitalization: implications for” Ref: 2670515, deals with an important subject. It is well written. Minor revisions are needed.

1. The main request for the author is addition of the following:

§  List of abbreviations mentioned throughout the study.

§  Previous publications describing the factors impacting COVID-19 incidence should be highlighted in the introduction section.

2. The topic of this article seems good contribution in the field with sounding output. This is why it is recommended for publication after limited revisions.

3. The study is useful for researches interested in optimizing an appropriate technique/methodology for detecting COVID-19 infection so, can afford good cure and recovery. The output of the mentioned methodology seems more accessible than other recommended expensive techniques.

4. The author himself mentioned that other parameters (BMI, CRP, and risk of death) should be reconsidered in future similar studies.

5. The cited references seem sufficient for this study as mentioned in the report sent due to this study.

Reviewer 3 Report

Comments and Suggestions for Authors

Thank you for the opportunity to review this manuscript. In order to enhance its suitability for publication in the journal, major revisions are needed considering the following items:

1.More recent research concerning the use of CRP as a predictor for COVID-19 mortality and disease severity should be included into the introduction section

2.Considering that the primary outcome of the study is in-hospital mortality and the objective is to analyze the CRP value in correlation with this outcome, why do the authors use the expressions "died from COVID-19", "risk of death due to COVID-19", " death in the hospital from COVID-19", "COVID-19 outcome"? COVID-19 is not a predictor in the logistic regression model, instead it is a "characteristic" of all the patients included in the study. All these expressions should be replaced with the primary outcome, as it is described in the methods section.

3.The Authors should revise the following phrase from the results section: ” Patients with MAX CRP levels > 10 mg/dL were four times more likely to die from COVID-19 compared to patients with MAX CRP levels ≤ 10 mg/dL, whereas patients with initial CRP values > 10 mg/dL were only 62% more likely to die than patients with initial CRP values ≤ 10 mg/dL.” . The comparation by % is confusing.

4.The results are a little hard to read because the authors have elaborated extensively on all the details, which are already very clearly shown in the logistic regression model. Due to this, the essence of the results is hidden in the background, namely what brings value and novelty to this study. You should find a way to highlight your strong points.

5.In the discussion section, there are not enough explanations related to the second outcome, which is clinical severity based on the American Thoracic Society Group (ATSG) criteria. The authors should revise this and clearly specify which of the outcomes are they addressing. Consistency in the terminology within the manuscript is crucial. It's important for the reader to have a clear understanding of the specific outcome under discussion at all times. In the case of "hospitalization outcome," it lacks clarity as to whether it pertains to the outcome of mortality or the severity of the disease.

6.The Authors should revise this part of the conclusion: ” Indeed, for patients ≥60, BMI < 25 kg/m2 was associated with increased risk of death, whereas the reverse was true for patients <60”, as the sentence above this phrase implies that age has a major impact on BMI, not vice versa. Therefore, it should emphasize that among patients with a BMI less than 25 kg/m2, age > 60 years old was linked to a higher in-hospital mortality rate (considering that there is already an explanation in the discussion section that obesity is less common in patients over 60 years old).

In addition to the aforementioned points, I believe that the study's concept is highly commendable and pragmatic. It has the potential to significantly enhance clinical practice, particularly given the widespread accessibility of this cost-effective laboratory analysis in all healthcare facilities treating COVID-19 patients.

Round 2

Reviewer 3 Report

Comments and Suggestions for Authors

All comments have been successfully addressed. Congratulations on the diligent effort put into this work. The manuscript is now highly suitable for publication in its current state.

As for comment number 4, I don't believe it's essential to employ bold formatting within the table (I even mentioned the tables' clarity). Instead, my concern was directed at the textual information explaining the results. However, with all the manuscript revisions made, it now appears much more straightforward to comprehend.